# Genomic Analysis of the Glutathione *S*-Transferase Family in Pear (*Pyrus communis*) and Functional Identification of *PcGST57* in Anthocyanin Accumulation

**DOI:** 10.3390/ijms23020746

**Published:** 2022-01-11

**Authors:** Bo Li, Xiangzhan Zhang, Ruiwei Duan, Chunhong Han, Jian Yang, Long Wang, Suke Wang, Yanli Su, Lei Wang, Yongfei Dong, Huabai Xue

**Affiliations:** 1Zhengzhou Fruit Research Institute, Chinese Academy of Agricultural Sciences, Zhengzhou 450009, China; nicelibo2020@163.com (B.L.); zhangxiangzhan@caas.cn (X.Z.); weiwei6182021@163.com (R.D.); hanchdoudou@163.com (C.H.); yangjian@caas.cn (J.Y.); wanglong02@caas.cn (L.W.); wangsuke@caas.cn (S.W.); suyanli@caas.cn (Y.S.); ewlei@163.com (L.W.); 18300702755@163.com (Y.D.); 2College of Horticulture and Plant Conservation, Henan University of Science and Technology, Luoyang 471023, China

**Keywords:** pear (*Pyrus communis*), glutathione *S*-transferase, anthocyanins, *PcGST57*

## Abstract

Anthocyanin accumulation in vacuoles results in red coloration in pear peels. Glutathione *S*-transferase (GST) proteins have emerged as important regulators of anthocyanin accumulation. Here, a total of 57 PcGST genes were identified in the European pear ‘Bartlett’ (*Pyrus communis*) through comprehensive genomic analysis. Phylogenetic analysis showed that PcGST genes were divided into 10 subfamilies. The gene structure, chromosomal localization, collinearity relationship, *cis*-elements in the promoter region, and conserved motifs of PcGST genes were analyzed. Further research indicated that glutamic acid (Glu) can significantly improve anthocyanin accumulation in pear peels. RNA sequencing (RNA-seq) analysis showed that Glu induced the expression of most PcGST genes, among which *PcGST57* was most significantly induced. Further phylogenetic analysis indicated that *PcGST57* was closely related to GST genes identified in other species, which were involved in anthocyanin accumulation. Transcript analysis indicated that *PcGST57* was expressed in various tissues, other than flesh, and associated with peel coloration at different developmental stages. Silencing of *PcGST57* by virus-induced gene silencing (VIGS) inhibited the expression of *PcGST57* and reduced the anthocyanin content in pear fruit. In contrast, overexpression of *PcGST57* improved anthocyanin accumulation. Collectively, our results demonstrated that *PcGST57* was involved in anthocyanin accumulation in pear and provided candidate genes for red pear breeding.

## 1. Introduction

Pear is one of the most cultivated fruit species worldwide for its economic benefit and valuable nutrition [1]. It has been cultivated for more than 3000 years, with an annual world production of approximately 23.9 million tons (FAOSTAT 2019). As an important commercial characteristic, pears with a red pericarp are more favored by consumers [2]. Developing red-colored cultivars has been an important target of pear-breeding programs due to the growing demands for high-quality pear fruit.

Anthocyanin accumulation in vacuoles results in red coloration in different tissues, including flowers, leaves, and pericarps. Anthocyanins act as important attractants or repellants to insects and animals [3]. Moreover, anthocyanins are involved in resistance to biotic and abiotic stress, including pathogen damage, ultraviolet radiation, and low temperature [4,5]. Additionally, anthocyanins serve as an important source of antioxidant compounds, which are responsible for human health and contribute to the economic value of horticultural products [5,6,7].

Anthocyanins are produced starting from phenylalanine in the cytoplasm through the flavonoid biosynthetic pathway [8], which involves the catalytic reaction of enzymes encoded by *PAL*, *C4H*, *4CL*, *CHS*, *CHI*, *F3H*, *F3′H*, *DFR*, *ANS*, *UFGT*, and other structural genes [9]. The expression of these structural genes is regulated by the MYB-bHLH-WD40 (MBW) regulatory complex, which consists of R2R3-MYB transcriptional factors, bHLH, and WD40 proteins [10]. Anthocyanins are transported to the vacuole by anthocyanin transporters [11]. Currently, three different mechanisms of flavonoid transportation have been proposed, namely vesicle trafficking, membrane transporters, and glutathione *S*-transferase (GST) mediated transporters [12,13]. These mechanisms function in coordination and cooperate with each other [13]. Cytoplasmic vesicle-like structures containing anthocyanins have been observed to fuse into the vacuole in lisianthus petals (*Eustoma grandiflorum*) [14] and *Arabidopsis* [15]. Members of the plant MATE family proteins were involved in flavonoid/H^+^ exchange and acylated anthocyanin transportation [13,16]. An ABCC-type transporter (*ZmMrp3*) is required for anthocyanin transport to the vacuole in *Zea mays* [17], and two-fruit specificity anthoMATE1 (AM1) and AM3 proteins were also investigated in grapevine [18]. Recently, several studies revealed the involvement of GST genes in anthocyanin accumulation [11,19,20,21,22]. GST can combine with flavonoids to form a GST-flavonoid complex to protect flavonoids from oxidation or guide them into the central vacuole [23].

Glutathione *S*-transferases (GSTs) are a superfamily that encodes multifunctional enzymes, which recruit the tripeptide glutathione as a coenzyme or co-substrate to participate in cell activities [24,25]. GST genes have been identified in many species and function in various aspects of plant growth and development, including detoxification of xenobiotic [26,27], biotic and abiotic stress responses [28,29], and secondary metabolism [25]. Recent studies have indicated that GST genes play important roles in plant anthocyanin accumulation. It was first demonstrated in *Zea mays* that *ZmBZ2* encodes a GST protein, which is involved in the transportation of anthocyanins to vacuoles [30]. Subsequently, studies of GST genes related to anthocyanin transportation were reported in many other species, including *PhAN9* in petunia [31], *AtTT19* in *Arabidopsis* [32], *CkmGST3* in cyclamen [33], *VviGST4* in grape [34], *PpGST1* in peach [20], *FvRAP* in strawberry [21,35], and *MdGSTF6* in apple [36]. Although GST is involved in anthocyanin accumulation in many species, information on GST in European pear and the role of GST genes in pear coloration is still lacking.

In this study, a total of 57 GST genes were identified in the genome of the ‘Bartlett’ pear and divided into 10 subfamilies. The phylogenetic evolution, syntenic relationships, gene structure, chromosome location, protein domain, *cis*-elements, and conserved motifs of PcGST genes were comprehensively analyzed. A *PcGST57* gene was selected based on RNA sequencing (RNA-seq) analysis, sequence alignment, and phylogenetic analysis. The transcript patterns of *PcGST57* in various pear cultivars, tissues, and different developmental stages were analyzed. Transient expression analysis indicated that *PcGST57* overexpression improved pear coloration, while *PcGST57* silencing reduced anthocyanin accumulation. These results provide a systematic analysis of the pear GST gene family and provide candidate GST genes for anthocyanin accumulation, thereby improving red pear breeding.

## 2. Results

### 2.1. Identification and Sequence Characterization of PcGST Family Members

To identify the GST family genes in the ‘Bartlett’ DH pear genome, an HMM search was conducted based on the HMM profiles (PF00043 and PF02798) from the Pfam database, and a local BLASTP search was performed using 63 *Arabidopsis* GST protein sequences as queries [36]. The ‘Bartlett’ DH was derived from the double haploid pear ‘Bartlett’ as described by Bouvier et al. [37]. A total of 78 candidate protein sequences were obtained with E-values under the threshold of 10^−5^. To further examine the completeness of the conserved domain, the 78 candidate proteins were searched in the NCBI Conserved Domain Search and Pfam database. Finally, a total of 57 PcGST genes (named *PcGST1*–*PcGST57*) were identified in the ‘Bartlett’ DH pear genome. The 57 PcGST genes were renamed based on their chromosomal location (Table 1).

The coding sequence (CDS), protein length, isoelectric point (p*I*), and molecular weight (MW) of PcGST members were analyzed (Table 1). In the ‘Bartlett’ DH pear genome, PcGST12 was predicted to be the smallest GST protein with 100 amino acids (aa) and 11.6 kDa, and PcGST22 was the largest with 786 aa and 91.2 kDa. The p*I* of GST proteins varied from 4.92 (PcGST27) to 9.75 (PcGST46).

### 2.2. Phylogenetic Analysis of the PcGST Family

To clarify the phylogenetic relationship of PcGST proteins, a neighbor-joining (NJ) tree, comprising 63 AtGST proteins and 57 PcGST proteins, was constructed using MEGAX software (Figure 1). The PcGST family was divided into 10 subfamilies, including lambda (L), dehydroascorbate reductase (DHAR), metaxin (M), tetrachlorohydroquinone dehalogenase-like (TCHQD), glutathionyl hydroquinone reductase (GHR), γ-subunit of the eukaryotic translation elongation factor 1B (EF1Bγ), Zeta (Z), Theta (T), Phi (F), and Tau (U). These were consistent with the GST subfamily in other species, including tomato, strawberry, and apple [21,28,36]. Among these subfamilies, the Tau subfamily contained the most PcGST genes (24), accounting for more than half of the total PcGST genes, followed by 11 PcGST genes in the Phi subfamily. The TCHQD subfamily was the smallest subfamily with one PcGST gene.

### 2.3. Chromosomal Location and Collinearity Analysis of PcGST Genes

The chromosomal location of 57 PcGST genes was analyzed based on the annotations of the ‘Bartlett’ pear genome, and the PcGST genes were unevenly distributed on the 17 pear chromosomes (Figure 2). Chromosome 5 contained the most PcGST genes with 10 GST members, followed by six PcGST genes on chromosome 15. There was only one PcGST gene located on chromosomes 1, 7, 11, and 12.

The collinearity of PcGST genes was analyzed, and a total of 15 segmental duplication events were identified, involving 26 of the PcGST genes (Figure 3A). *PcGST1*, *PcGST10*, *PcGST13*, and *PcGST40* were involved in multiple repetitive events (Appendix A). Two or more genes located within 20 kb were defined as gene clusters [38]. A total of 11 PcGST gene clusters were identified, involving 18 genes, such as *PcGST10/11* on chromosome 4 and *PcGST17/18* on chromosome 5. Three linear GST genes were identified on chromosomes 2 (*PcGST3/4/5*), 3 (*PcGST6/7/8*), and 5 (*PcGST13/14/15*), indicating that they originated from continuous replication events.

To further analyze the evolutionary and collinearity relationship of PcGST genes, synteny analyses were conducted using the orthologous GST gene pairs among *Arabidopsis*, pear, and apple. A total of 22 PcGST genes had a homologous relationship with *Arabidopsis*, involving 33 pairs of homologous genes. A total of 42 PcGST genes had a homologous relationship with apple, involving 80 pairs of homologous genes (Figure 3B). Some collinear gene pairs appeared only in *Arabidopsis* or apple gene pairs (Appendix A). There were 38 collinear gene pairs, involving 22 PcGST genes that were only detected between apple and pear, such as *PcGST2/MD02G1204200*, which was not found in *Arabidopsis*. In addition, 14 PcGST genes had no collinearity relationship with *Arabidopsis* and apple.

### 2.4. Structure and Motif Analysis of PcGST Genes

The MEME motifs search was performed to analyze the conservative motifs in PcGST genes, and a total of 10 conservative motifs were identified (Figure 4A). Motif 3 existed in all subfamilies, indicating that the motif is conserved in the GST proteins. Motifs 5, 6, and 8 only existed in the Tau subfamily, suggesting that they might be the specific motifs of the Tau subfamily. Motifs 1, 2, 5, and 7 appeared in the Tau subfamily, except in *PcGST47*. Motifs 1 and 4 were present in all members of the Phi subfamily. The GST family members of the same subfamily had similarities in exon/intron structures and conservative motifs, which supports the reliability of subgroup classification.

To elucidate the structural characteristics of the PcGST genes, the gene structure of the PcGST family was analyzed (Figure 4B). The exon number of most PcGST genes ranged from two to 10, with wo exceptions: *PcGST47* with 11 exons and *PcGST22* with 21 exons. Most PcGST genes in the same subfamily showed a similar exon/intron structure. Among the plant-specific GST genes in the Phi subfamily, *PcGST6* had two exons; *PcGST39*, *PcGST28*, *PcGST32*, *PcGST54*, *PcGST57*, and *PcGST23* had three exons, and *PcGST7*, *PcGST8*, *PcGST9*, and *PcGST45* had four exons. In the Tau subfamily, most GST members contained two exons, except for *PcGST14*, *PcGST36*, *PcGST15*, *PcGST51*, *PcGST19*, and *PcGST47*, with exons ranging from three to 11.

### 2.5. Cis-Element Analysis of PcGST Promoters

To understand the potential regulation mechanism of PcGST genes, *cis*-elements in the promoter regions of PcGST genes were analyzed. Various types of *cis*-elements were identified (Figure 5), including defense and stress responsive elements (e.g., W-box, ARE element, TC-rich repeats, GC-motif, and LTR element), hormone-related responsive elements (e.g., ABRE, GARE-motif, P-box, TATC-box, AuxRE, AuxRR-core, TGA-box, TGA-element, ERE, TCA-element), and light responsive elements (e.g., ACE, G-box, GT1-motif, MRE), indicating the possible function of GST members. Additionally, many MYB binding site elements involved in flavonoid biosynthesis were identified, which might be related to the promotion of anthocyanin accumulation.

### 2.6. Expression Analysis of the PcGST Family by Glutamic Acid (Glu)-Induced Coloration

Our previous study revealed that an appropriate concentration of Glu can promote anthocyanin accumulation in the red blush pear peel. To reveal the underlying mechanism, ‘Danxiahong’ pears were treated with 0.06% Glu or sterilized water and then transferred to a light incubator at 17 °C with continuous light. The peels were sampled at indicated time points, namely 0, 3, 12, 24, and 72 h, for RNA-seq analysis.

RNA-seq analysis demonstrated that the expression of most GST genes, structural genes and some regulatory genes, which were associated with anthocyanin accumulation, were significantly upregulated following Glu treatment. Increasing studies have suggested that GST proteins are involved in anthocyanin accumulation in many plant species. Thus, all PcGST members were analyzed based on RNA-seq. The results revealed that 53 PcGST genes were expressed in at least one period, and four PcGST genes were not expressed at any time (Figure 6). Remarkably, most PcGST members of the Phi subfamily, which were associated with anthocyanin accumulation, exhibited increased expression levels. Notably, *PcGST57* in the Phi subfamily was significantly upregulated, with a fold change of more than 18, which was the highest at 72 h, indicating that *PcGST57* might play an essential role in anthocyanin accumulation.

### 2.7. Phylogenetic Analysis and Sequence Alignment of PcGST57 and Anthocyanin-Related GST Genes

To elucidate the relationship between *PcGST57* and GST genes as distinct from other species, a phylogenetic tree was constructed with *PcGST57* and GST genes associated with anthocyanin accumulation from different species, including apple [36], peach [20], *Arabidopsis* [32], grape [39], strawberry [35], and other plant species. *PcGST57* had the closest relationship with apple *MdGSTF6*, followed by peach *PpGST1* (Figure 7A). *MdGSTF6* and *PpGST1* were the highest targets of *PcGST57* in apple and peach by a BLASTP search, respectively. Sequence alignment indicated that *PcGST57* shared the same conserved domain with anthocyanin-related GST genes (Figure 7B), indicating that *PcGST57* may play an important role in anthocyanin accumulation, and it was selected as the candidate GST gene for further analysis.

### 2.8. Expression Patterns of PcGST57

To determine the expression levels of *PcGST57* in pears with different peel colors, pears, including ‘Zaosu’, ‘Mansoo’, ‘Danxiahong’, ‘Red Clapp’s Favorite’, and ‘Red Zaosu’ were selected for analysis. RT-qPCR analysis showed that the expression level of *PcGST57* was low in the green pear ‘Zaosu’ and gradually increased in the red blushed pear ‘Danxiahong’ and the full red pear ‘Red Clapp’s Favorite’ (Figure 8). Expression was highest in the purple–red pear ‘Red Zaosu’. The expression level in the russet pear ‘Mansoo’ was significantly lower than that in the other pears (Figure 8A). The peel color and expression level of *PcGST57* showed a positive correlation.

To examine the expression levels of *PcGST57* in different tissues, various tissues from pear ‘Danxiahong’ were sampled, including leaves, petals, anthers, floral shoots, and floral receptacles sampled at 0 DAF, and peels and flesh sampled at 100 days after full blossom (DAF). RT-qPCR analysis showed that *PcGST57* exhibited a low transcript level in the floral shoot, floral receptacle, anther, and petal (Figure 8B). The expression level in young leaves was 2.90–5.72 times that of the above tissues. Notably, *PcGST57* was most highly expressed in the peel, while the lowest expression level was observed in the flesh, indicating that *PcGST57* was specifically expressed in various pear tissues and possibly involved in anthocyanin accumulation in the peel.

The transcript levels of *PcGST57* in ‘Danxiahong’ pear peels at different developmental stages were analyzed. *PcGST57* exhibited a low expression level at early developmental stages (before 55 DAF). The expression level gradually increased and reached its highest point at more than 500-fold in the color transition period (115 DAF) compared to 10 DAF, and then declined at later stages (Figure 8C). Moreover, the expression of *PcGST57* in pear flesh at different developmental stages was also investigated. The transcript level of *PcGST57* showed no significant correlation with fruit coloring during pear development (Figure 8D). The specific expression of *PcGST57* indicated that *PcGST57* was associated with fruit coloring and anthocyanin accumulation.

### 2.9. Function Identification of PcGST57 in Anthocyanin Accumulation

To gain insight into the biological function of *PcGST57* in fruit coloring and anthocyanin accumulation, *PcGST57* was cloned from ‘Danxiahong’ pear. Sequence alignment indicated that the coding sequence of *PcGST57* in ‘Danxiahong’ was identical to that of ‘Bartlett’ DH genome, which has an open reading frame (ORF) of 648 bp encoding 215 amino acids. The coding sequence of *PcGST57* was inserted into the overexpressing vector pCAMBIA1302 and transformed into *Agrobacterium tumefaciens* strain GV3101. The *A. tumefaciens* strain containing recombinant vector was transiently overexpressed in ‘Danxiahong’ peel as described by Zhang et al. [40]. Fifteen days after injection, the transcript abundance of *PcGST57* in overexpressing pear peels (PcGST57-OE) was significantly higher than that of the empty vector (Figure 9C). Correspondingly, the pear peels of overexpressing *PcGST57* exhibited improved coloration compared to pears overexpressing the empty vector (Figure 9A). Physiological measurements showed that peels overexpressing *PcGST57* accumulated a higher total phenol and anthocyanin content than the empty vector (Figure 9D,E), indicating that *PcGST57* overexpression promoted anthocyanin accumulation in pear.

To further determine the biological function of *PcGST57* in anthocyanin accumulation, tobacco rattle virus (TRV) mediated virus-induced gene silencing (VIGS) technology was applied to silence *PcGST57* expression in ‘Danxiahong’ pear. The transient VIGS assay indicated that the PcGST57-TRV pear peel exhibited green pigmentation around the injection sites, while the pear peel injected with the empty vector showed normal pigmentation (Figure 9B). Meanwhile, the transcription level of *PcGST57* was significantly decreased in PcGST57-TRV pear peels compared to empty vector peels (Figure 9F). The measurement of anthocyanins and total phenol indicated that *PcGST57* silencing decreased fruit coloration (Figure 9G,H). Furthermore, the results suggested that *PcGST57* plays an important role in anthocyanin accumulation in pear coloration.

## 3. Discussion

GST genes were first reported in animals for their metabolism and detoxification effects of drugs [41]. Afterwards, GST was reported in plants for its ability to protect maize from herbicide effects [42]. Initially, only three GST subfamilies were identified, including Theta, Tau, and Zeta [43]. Subsequently, the GST gene family was systematically and comprehensively analyzed, and increasing GST members were identified in various plant species, including maize [42], soybean [44], *Arabidopsis* [45], apple [36], strawberry [21], tomato [28], *Citrus sinensis* [46], kiwifruit [47], cotton [19], radish [22], wheat [48], and pepper [49]. The number of GST family members and the number of subfamilies vary in different plant species. There are 61 GST genes involved in six subfamilies in *Citrus sinensis* [46], 90 GST genes involved in 10 subfamilies in tomato [28], and 69 GST genes involved in nine subfamilies in apple [36]. In this research, 57 PcGST genes were identified in European pear and were divided into 10 subfamilies. The different members and subfamilies might be caused by species specificity.

Whole-genome duplication, segmental duplication, and tandem duplication events were the main sources of expansion of new genes and gene families [50]. In this research, two manners of duplication events in PcGST genes were identified, and the number of segmental duplication events was more than tandem duplication event, which was consistent with SlGST genes in tomato [28]. Collinearity analysis revealed the evolutionary relationship among GST genes in different species. More than three-fifths of the PcGST genes had collinear relationships with apple GST genes, while less than half of the PcGST genes had collinear relationships with *Arabidopsis* GST genes. This may be due to the close genetic relationship between pear and apple, as they are both Rosaceae species. Some GST genes were only present in *Arabidopsis* or apple gene pairs, indicating that they might have been lost in their ancestors during evolution or may have been generated by species differentiation. Interestingly, some PcGST genes had multiple counterparts in apple, and these genes might have undergone different amplification processes during evolution.

GST genes in the same group with similar exon/intron structures and conserved motifs may have similar functions. *PcGST57* was clustered in the Phi family and grouped with *MdGSTF6*, *PpGST1*, and GST members associated with anthocyanin accumulation, which supports the accuracy of our analysis. Interestingly, some GST members within the same subfamily shared different conserved motifs and exon/intron structures, indicating specific functions. More than half of the PcGST genes contained no more than three exons, which was consistent with previous studies [28,51], indicating the conservation of the GST family during evolution.

The application of exogenous Glu has been shown to promote anthocyanin accumulation in ‘Fuji’ apple and peach leaves [52,53]. Exogenous Glu promoted peel coloring and total sugar accumulation in litchi [54]. Our research indicated that Glu could induce anthocyanin accumulation in pear. RNA-seq also revealed that most structural genes and partial regulatory genes were induced by Glu treatment (Figure 6), and most GST genes showed upregulation, suggesting the important role of the GST family in Glu-induced anthocyanin accumulation.

The GST genes in the same subfamily sharing the same conserved domain may have similar functions. The Tau subfamily, as the biggest GST subclass, was reported to be related to xenobiotic metabolism and detoxification [55]. The Phi subfamily is a plant-specific class, and most GST genes in the Phi subfamily were reported to be related to anthocyanin transportation. The Phi subfamily GST members in *Arabidopsis*, apple, peach, strawberry, and other species have been associated with anthocyanin accumulation. *PcGST57* exhibited a close relationship with *AtGSTF12* (*AtTT19*), which transports anthocyanins from the cytosol to the vacuole as a carrier protein [56]. In our research, *PcGST57* was clustered with most Phi subfamily members and participated in anthocyanin accumulation, which was consistent with the function of GST genes in other plants.

The expression of *PcGST57* was significantly induced by more than 18-fold under Glu treatment (Figure 6). In pear cultivars with different colors, the expression level of *PcGST57* showed a positive correlation with peel color, and *PcGST57* showed a significantly higher expression level in peels than in other tissues. Similarly, the expression of *PcGST57* increased in the early stage of fruit development, reaching the highest level in the fruit color transition period (more than 500-fold), and then gradually decreased but still maintained a high expression pattern (Figure 8). These results indicate that *PcGST57* might be vital in anthocyanin accumulation in peels rather than in other tissues.

Although our research showed that Glu could promote anthocyanin accumulation in red blush pear, it was unclear how plants respond to Glu signals to regulate the expression of *PcGST57* and other genes. Glu can be converted into intermediate substances, such as 5-aminolevulinic acid (ALA), which is the synthetic precursor of plant phytochromes, and can regulate anthocyanin synthesis [57]. Glu can also increase the sugar content in fruits, which could provide an abundant sugar base for anthocyanin synthesis [54,58]. In plants, many secondary metabolites are essential for targeting vacuoles for their phytotoxic characteristics, even for cells that produce them [59]. Anthocyanin retention in the cytoplasm is toxic to the cell and prevents the biosynthesis of new anthocyanins [30]. Some studies have shown that GST functions by directly binding to anthocyanins or transferring glutathione to anthocyanins to form glutathione *S*-conjugates [30,31]. Anthocyanin accumulation in the cytoplasm drives *PcGST57* to transport anthocyanins. This might be why pear peels respond to Glu signals to regulate the expression of *PcGST57*, affecting anthocyanin accumulation, but further experimental verification is required.

Most PcGST members were induced by Glu application and were upregulated in RNA-seq analysis. It has been reported in many plants that anthocyanin-related GST genes could be activated by MYB transcription factors. Overexpression of *PAP1* activated the expression of *TT19* in *Arabidopsis*, which was required for anthocyanin transportation [60]. LcMYB1 was involved in anthocyanin accumulation by activating the expression of *LcGST4* in litchi, and two MYB binding sites were found in the promoter region of *LcGST4* [8]. LhMYB12-lat could improve anthocyanin transportation by upregulating the transcriptional activity of *LhGST* and activating *LcGST* expression in lilies [11]. PpMYB10.1 could bind to the promoter and transactivate the transcription of anthocyanin-related *PpGST1* [20]. Ten MYB binding sites were identified in *PcGST57* (Figure 5), indicating that the expression of *PcGST57* might be regulated by MYB transcription factors, further affecting anthocyanin accumulation. Whether MYB transcription factors directly regulate *PcGST57* in pear involving anthocyanin accumulation remains to be further studied.

## 4. Materials and Methods

### 4.1. Plant Materials

The red blush pear ‘Danxiahong’ was planted in the orchard of Zhengzhou Fruit Research Institute, Chinese Academy of Agricultural Sciences, and was used as the experimental material. The green fruits of ‘Danxiahong’ pear that were not in direct sunlight were collected 20 days before full maturity on 30 July 2020 from six-year-old pear trees. The fruits were treated with distilled water or a 0.06% glutamic acid (Glu) solution and then incubated in a light incubator with continuous light and 80% relative humidity at 17 °C, as described by Bai et al. [61]. The peels were collected at indicated time points, including 0, 3, 12, 24, and 72 h for RNA extraction and RNA-seq analysis. RNA-seq reads were aligned to the *Pyrus communis* ‘Bartlett’ DH Genome v2.0 from Genome Database for Rosaceae (GDR, https://www.rosaceae.org/, accessed on 5 December 2020).

Tender leaves, floral shoots, floral receptacles, anthers, and petals were sampled at 0 day after full blossom (DAF) from the ‘Danxiahong’ pear tree. Young fruits were first collected at 10 DAF, and then the peels and flesh were collected every two weeks until fruit maturity. These samples were immediately frozen in liquid nitrogen and transferred to a −80 °C freezer for further RNA extraction and transcription analysis.

### 4.2. Identification and Chromosomal Location of Pear PcGST Genes

The genome data of pear (*Pyrus communis* ‘Bartlett’ DH Genome v2.0) and apple (*Malus domestica* GDDH13 v1.1) were downloaded from the Genome Database for Rosaceae (GDR, https://www.rosaceae.org/, accessed on 5 December 2020). *Arabidopsis* genome data was downloaded from the *Arabidopsis* Information Resource (TAIR, https://www.arabidopsis.org/index.jsp, accessed on 5 December 2020). Hidden Markov Model (HMM) profiles (PF00043 and PF02798) were obtained from the Pfam database (http://pfam.xfam.org/, accessed on 18 January 2021), and the PcGST candidate genes were identified using HMMER 3.0 software. Genes obtained by the BLAST program with E-values under the threshold of 10^−5^ were used for further filter analysis. The online CD search (https://www.ncbi.nlm.nih.gov/Structure/cdd/wrpsb.cgi, accessed on 3 February 2021) and Pfam search were used to examine the GST domain and further identify candidate GST genes.

The chromosomal location of PcGST genes was obtained from the ‘Bartlett’ pear genome in the GDR database. PcGST genes were mapped to pear chromosomes by MapInspect software (https://mapinspect.software.informer.com, accessed on 12 February 2021).

### 4.3. Phylogenetic and Evolutionary Analysis of Pear PcGST Genes

The GST protein sequences of pear and *Arabidopsis* were aligned using the Clustal X version 2.0 program. A phylogenetic tree was constructed to analyze the evolutionary relationships of the GST genes using the neighbor-joining method with 1000 bootstrap by MEGAX [38].

The Multiple Collinearity Scan toolkit (MCScanX) was used for collinear analysis, and the replication patterns of PcGST genes were analyzed according to the method of Wang et al. [62]. The collinear relationship of the PcGST genes was visualized using the advanced circus and multiple synteny plot packages in TBtools [63].

### 4.4. Gene Structure and Cis-Element Analysis of PcGST Members

The conserved domain and exon/intron structure organization of PcGST genes were determined by Gene Structure Display Server 2.0 (GSDS: http://gsds.cbi.pku.edu.cn/, accessed on 2 March 2021). The conserved motifs of PcGST proteins were determined by MEME online software version 5.4.1 (https://meme-suite.org/meme/tools/meme, accessed on 15 March 2021). The gene structure of the PcGST genes was visualized using the gene structure view package in TBtools.

For *cis*-element analysis, a 2000-bp sequence 5′ upstream from the start codon of the PcGST gene was extracted from the *Pyrus communis* ‘Bartlett’ DH Genome by TBtools [63], and the *cis*-acting elements were identified by the online database PlantCARE (http://bioinformatics.psb.ugent.be/webtools/plantcare/html/, accessed on 3 April 2021).

### 4.5. RNA Isolation and Transcript Analysis

Total RNA was extracted from the pear peel using an RNA Extraction Kit (Zoman, Beijing, China) according to the manufacturer’s protocol. The genomic DNA was removed by DNase I (Zoman, Beijing, China), and the first-strand cDNA was synthesized using TransScript One-Step gDNA Removal and cDNA Synthesis SuperMix (TransGen Biotech, Beijing, China). RT-qPCR was performed with TransStart Top Green qPCR SuperMix (TransGen Biotech, Beijing, China) using the Roche LightCycler 480 system (Roche, Basel, Switzerland) according to the manufacturer’s instructions. The pear *PcTubulin* gene was used as an internal control for gene expression analysis. Transcript levels were analyzed using the 2^−ΔΔCt^ method [64]. The primers used for RT-qPCR are listed in Appendix A. The heatmap package of TBtools was conducted to analyze the expression level of PcGST genes based on the RNA-seq data.

### 4.6. Functional Characterization of PcGST57 by Transient Expression

For the transient overexpression assay, the full-length coding sequence of *PcGST57* was inserted into the multiple cloning site (MCS; Nco*I*) of the pCAMBIA1302 vector driven by the cauliflower mosaic virus (CaMV) 35S promoter (PcGST57-OE). For the virus-induced gene silencing (VIGS) assay, the specific 300 bp *PcGST57* fragment was inserted into the MCS (EcoR*I* and Kpn*I*) of the pTRV2 VIGS vector to generate the PcGST57-VIGS vector. The recombinant plasmids were separately transformed into *Agrobacterium tumefaciens* strain GV3101 cells. Needle-free syringes were used to inject the ‘Danxiahong’ pear fruit before the color transition period with the transformed *A. tumefaciens* cells containing recombinant or empty vectors. The fruits were sampled for phenotypic observation, anthocyanin and phenol determination, and gene expression analysis for 15 days after injection. At least 30 fruits were injected for each strain with the corresponding vector, and the transient expression assays were repeated three times.

### 4.7. Measurement of Anthocyanin and Total Phenol Content

Anthocyanins were extracted from the peels of injected pear fruits, and the anthocyanin content was measured as described by Wang et al. [52] with minor modifications. Approximately 200 mg of peels were frozen in liquid nitrogen and ground into powders, then 1 mL methanol solution containing 1% (*v*/*v*) HCl was added in the tube and incubated at 4 °C in the dark for 1 h. The tubes were centrifuged at 9500× *g* at 4 °C for 10 min, then the supernatant was transferred to a new tube for anthocyanin measurement. The absorbance was measured at 530 and 600 nm with SpectraMax i3x Multi-Mode Detection Platform (Molecular Devices, USA).The total phenol content was determined as described by Xu et al. [65].

### 4.8. Statistical Analysis

Experiments in the research were repeated for at least three times, the data exhibited as the mean ± SD from three independent replicates. Significant differences were analyzed by Student’s *t*-test, and the significance between corresponding controls was labeled with * (*p* < 0.05) or ** (*p* < 0.01).

### 4.9. Accession Numbers

Sequence data from this work can be found in the NCBI database (BioProject ID: PRJNA781276).

## 5. Conclusions

We identified 57 GST genes in the *Pyrus communis* ‘Bartlett’ DH genome and divided them into 10 subfamilies. RNA-seq analysis showed that *PcGST57* was significantly induced by Glu treatment. *PcGST57* was closely related to *MdGSTF6* and *PpGST1* and other GST genes, which were involved in anthocyanin accumulation in other species. *PcGST57* was associated with peel coloration rather than other tissues. Transient overexpression and silencing of *PcGST57* in ‘Danxiahong’ pear improved and decreased coloration, respectively. The results indicated that *PcGST57* is involved in anthocyanin accumulation in pear.

## Figures and Tables

**Figure 1 ijms-23-00746-f001:**
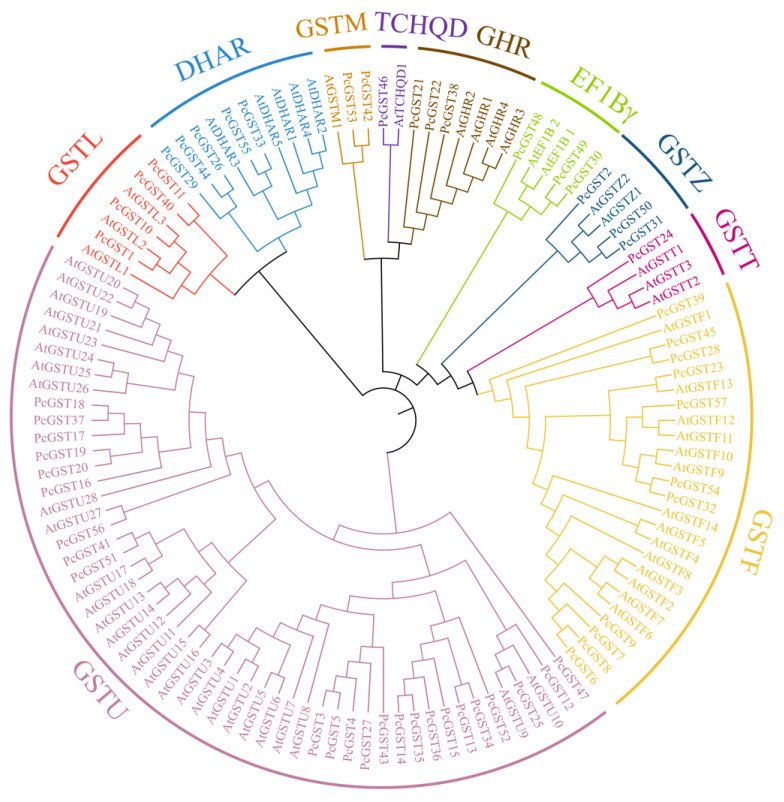
Phylogenetic analysis of GST genes from European pear and *Arabidopsis*. An unrooted phylogenetic tree was constructed using the full-length protein sequences of GST by MEGAX using neighbor-joining method, with 1000 bootstrap replicates. The branches with different colors indicates 10 subfamilies of GST proteins.

**Figure 2 ijms-23-00746-f002:**
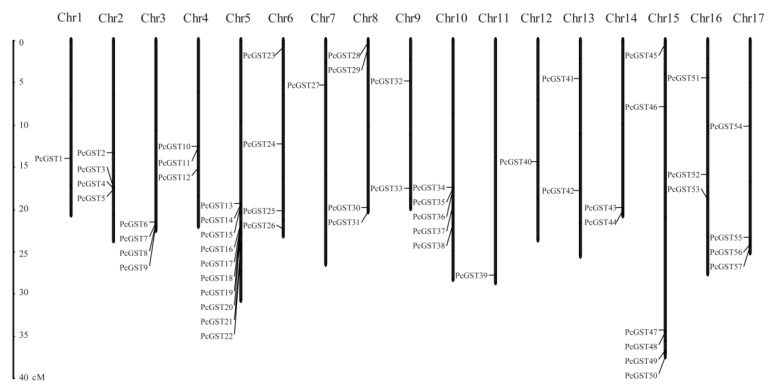
The chromosomal locations of PcGST genes in European pear. The position of PcGST genes was mapped to pear chromosomes based on the location information obtained from the genome of ‘Bartlett’ DH.

**Figure 3 ijms-23-00746-f003:**
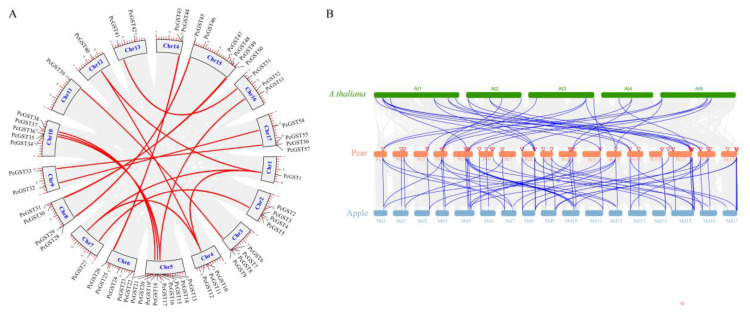
Collinearity analysis of the pear PcGST gene family. (**A**) Collinearity relationships of PcGST genes in pear genome. The panel exhibited 17 pear chromosomes in a circle with red lines connecting PcGST genes with WGD/segmental duplication events. Chromosome numbers were indicated on the circle with blue color. The PcGST genes were mapped to the chromosomes outside the circle. (**B**) Collinearity relationships of PcGST genes in pear, *Arabidopsis*, and apple. The blue lines indicated the PcGST genes with homologous relationship with *Arabidopsis* or apple.

**Figure 4 ijms-23-00746-f004:**
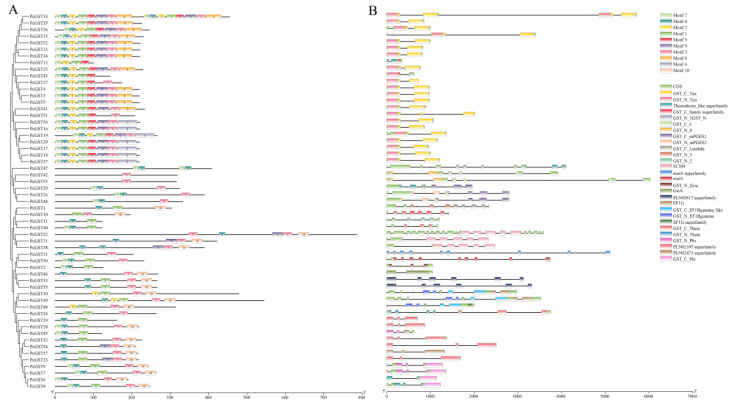
Structure and conserved domain analysis of PcGST genes. (**A**) Phylogenetic relationship and conserved motif analysis of PcGST genes. The boxes with different colors indicated different motifs. (**B**) Exon–intron structure and conserved domain analysis of PcGST genes. The conserved GST domains were exhibited with indicated colors, and the introns were shown as black lines.

**Figure 5 ijms-23-00746-f005:**
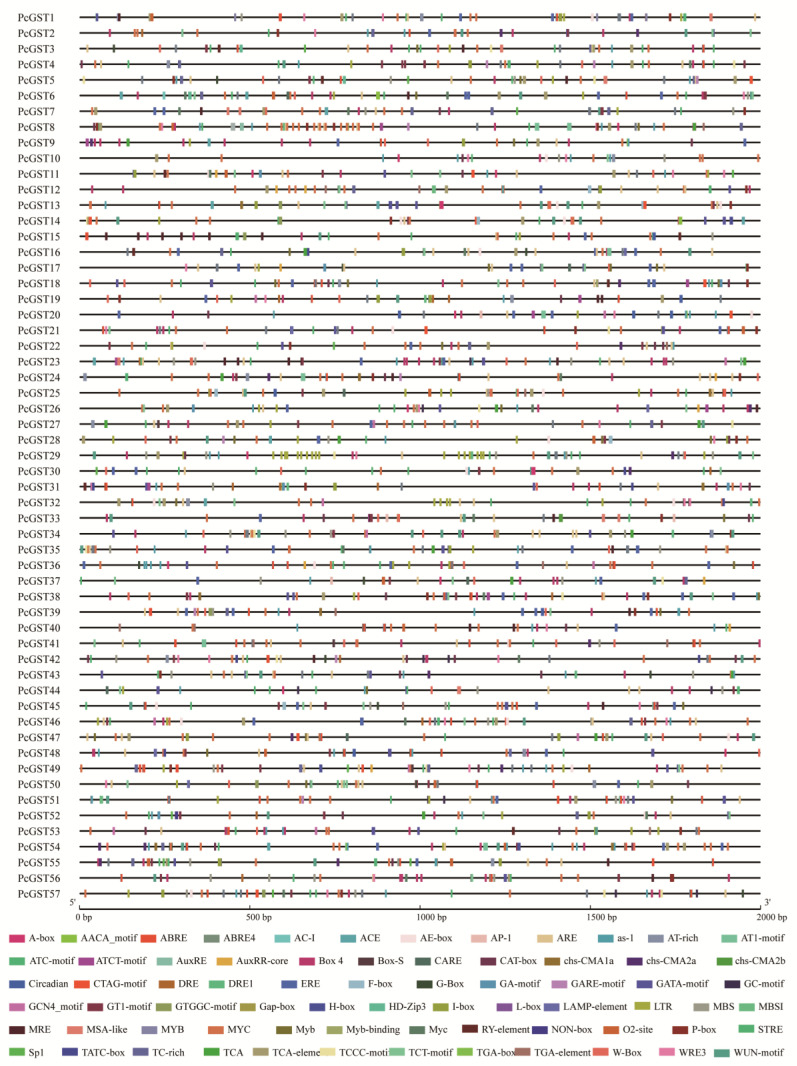
*Cis*-elements analysis in the promoter region of PcGST genes. The 2-kb of 5′ flanking sequence upstream from the start codon was obtained for analysis, and the boxes with different colors indicated different *cis*-elements.

**Figure 6 ijms-23-00746-f006:**
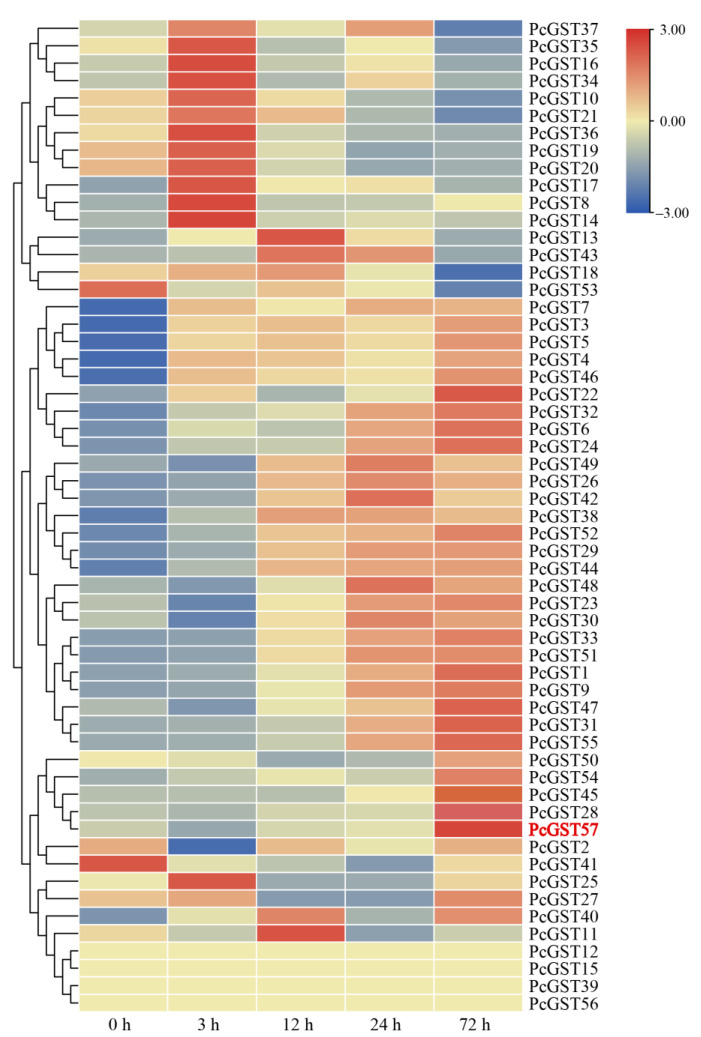
Heatmap of transcript profiles of PcGST genes in response to Glu treatment at indicated time points. The transcript abundance of PcGST genes was represented by different colors. The red and blue color on the panel indicate high and low expression, respectively. The data was obtained from three biological replicates.

**Figure 7 ijms-23-00746-f007:**
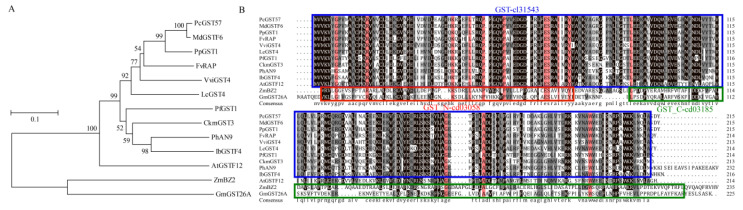
Phylogenetic analysis and sequence alignment of PcGST57 and its paralogs. (**A**) Phylogenetic relationships of PcGST57 paralogs was shown by phylogenetic tree. GST proteins from *Pyrus communis*, *Malus domestica*, *Arabidopsis*, *Prunus persica*, *Fragaria ananassa*, and several other species were used for the construction of a neighbor-joining tree. (**B**) Sequence alignment of PcGST57 and its paralogs. The GST domain with accession number of cl31543 was indicated by blue box, the red box indicated as GST-N domain with accession number of cd03058, and green box indicated as GST-C domain with accession number of cd03185. The accession number of GST members used in the analysis were as follows, MdGSTF6 (MD17G1272100), PpGST1 (Prupe.3G013600.1), FvRAP (gene31672), VviGST4 (AAX81329), LcGST4 (ALY05893), PfGST1 (BAG14300), CkmGST3 (BAM14584), PhAN9 (CAA68993), IbGSTF4 (MG873448), AtGSTF12 (AED92398), ZmBZ2 (AAA50245), and GmGST26A (NP_001238439).

**Figure 8 ijms-23-00746-f008:**
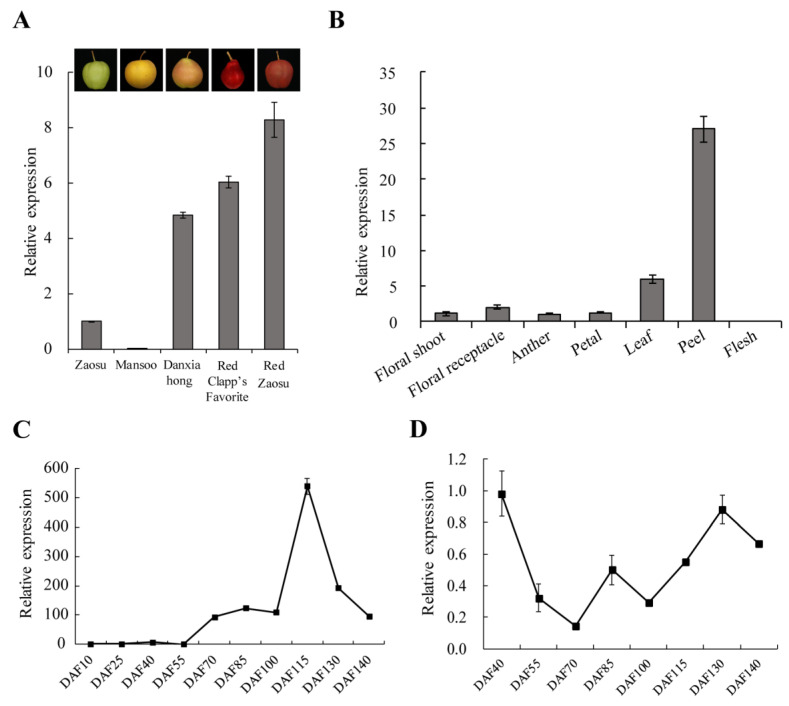
The transcript patterns of *PcGST57*. (**A**) Transcript analysis of *PcGST57* in different pears. Pears, including ‘Zaosu’, ‘Mansoo’, ‘Danxiahong’, ‘Red Clapp’s Favorite’, and ‘Red Zaosu’, with different colors were selected for analysis. (**B**) Transcript analysis of *PcGST57* in various pear tissues of ‘Danxiahong’. Various pear tissues of ‘Danxiahong’, including floral shoot, floral receptacle, leaf, petal, and anther at 0 DAF, and peel, flesh of 100 DAF were sampled for transcript analysis. (**C**) Transcript analysis of *PcGST57* in pear peels of ‘Danxiahong’ at different developmental periods. (**D**) Transcript analysis of *PcGST57* in pear flesh of ‘Danxiahong’ at different developmental periods. *PcTubulin* was used as the internal control for gene expression analysis. All the data indicate means ± SD of three biological replicates.

**Figure 9 ijms-23-00746-f009:**
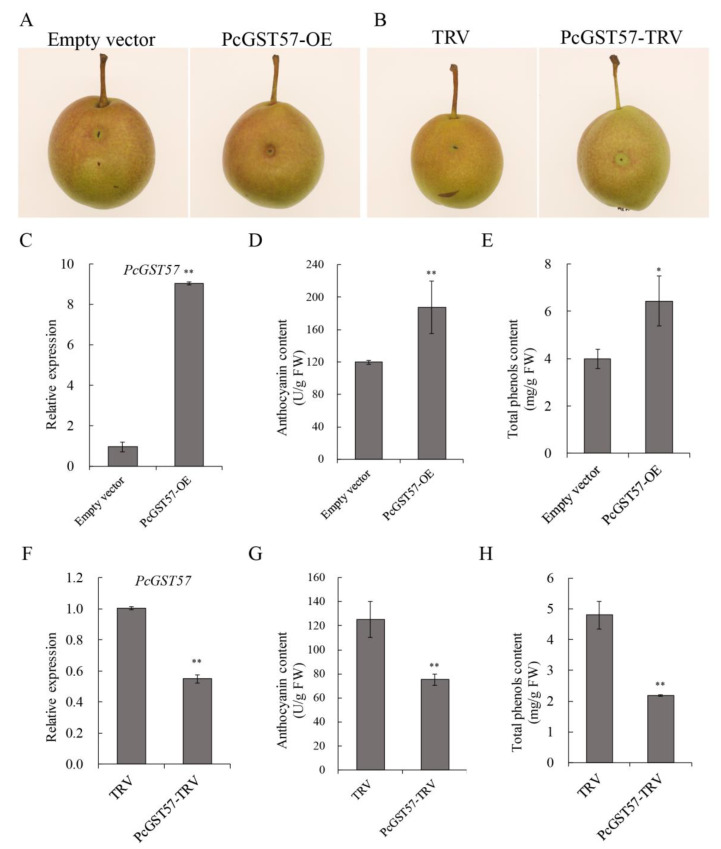
Functional identification of *PcGST57* by transient expression in ‘Danxiahong’ pear peels. Transient overexpression (**A**) and TRV-mediated silencing (**B**) assays of *PcGST57* in ‘Danxiahong’ pear peels. Transcript analysis of *PcGST57* in overexpressing peels (**C**) and silencing peels (**F**). Measurement of anthocyanin content in *PcGST57* overexpressing peels (**D**) and silencing peels (**G**). Measurement of total phenol content in *PcGST57* overexpressing peels (**E**) and silencing peels (**H**). The photograph were captured 15-day post injection, and the peels around the injection sites were sampled for expression analysis and measurement of anthocyanin content and total phenol content. *Pctubulin* was used as the internal control for gene expression analysis. All the data indicate means ± SD of three biological replicates. Asterisks indicate statistical significance (*, *p* < 0.05; and **, *p* < 0.01) determined by student’s *t*-test compared with corresponding controls.

**Table 1 ijms-23-00746-t001:** Molecular characteristics of the GST family genes identified in the ‘Bartlett’ DH pear genome.

Gene Name	Gene ID	Chromosome Location	CDS Length (bp)	Amino Acid Length (aa)	Isoelectric Point (p*I*)	Molecular Weight (kDa)
PcGST1	pycom01g14390	Chr1:14570338-14572677	912	303	8.73	34.14
PcGST2	pycom02g16660	Chr2:13905061-13906109	375	124	6.04	13.86
PcGST3	pycom02g20280	Chr2:18414352-18415328	660	219	5.21	25.23
PcGST4	pycom02g20290	Chr2:18422356-18423333	660	219	5.19	25.04
PcGST5	pycom02g20300	Chr2:18435026-18436003	660	219	5.45	25.27
PcGST6	pycom03g22330	Chr3:22462862-22464001	573	190	6.66	21.33
PcGST7	pycom03g22340	Chr3:22484661-22486019	792	263	8.48	29.48
PcGST8	pycom03g22350	Chr3:22495188-22496419	741	246	9.38	27.94
PcGST9	pycom03g22370	Chr3:22518852-22520127	732	243	9.11	27.34
PcGST10	pycom04g10330	Chr4:13096341-13097762	588	195	5.82	22.40
PcGST11	pycom04g10340	Chr4:13102770-13103977	366	121	9.46	13.75
PcGST12	pycom04g12760	Chr4:15730309-15730661	303	100	9.10	11.60
PcGST13	pycom05g16920	Chr5:20182936-20183765	666	221	5.27	25.34
PcGST14	pycom05g16930	Chr5:20184941-20190656	1365	454	6.12	51.96
PcGST15	pycom05g16940	Chr5:20190776-20194179	690	229	5.96	26.86
PcGST16	pycom05g19570	Chr5:22475360-22476224	663	220	6.14	25.58
PcGST17	pycom05g19590	Chr5:22500115-22501077	660	219	5.28	25.27
PcGST18	pycom05g19600	Chr5:22511832-22512833	660	219	5.27	25.39
PcGST19	pycom05g19620	Chr5:22528365-22529724	798	265	5.68	30.74
PcGST20	pycom05g19630	Chr5:22532945-22534113	660	219	5.64	25.50
PcGST21	pycom05g22990	Chr5:25074385-25076708	1266	421	8.27	48.07
PcGST22	pycom05g23030	Chr5:25095525-25099112	2361	786	9.04	91.22
PcGST23	pycom06g01020	Chr6:1015178-1016874	654	217	6.36	24.45
PcGST24	pycom06g07800	Chr6:12765956-12769705	792	263	9.45	29.96
PcGST25	pycom06g17080	Chr6:21128796-21129570	687	228	5.86	26.05
PcGST26	pycom06g20360	Chr6:23436022-23438824	1167	388	8.90	42.74
PcGST27	pycom07g06270	Chr7:5539987-5540718	522	173	4.92	19.68
PcGST28	pycom08g00510	Chr8:453769-454646	657	218	6.34	24.67
PcGST29	pycom08g00850	Chr8:661714-663672	975	324	8.94	36.30
PcGST30	pycom08g21180	Chr8:20733707-20736680	1437	478	8.27	54.21
PcGST31	pycom08g21730	Chr8:21146282-21151390	609	202	6.10	22.84
PcGST32	pycom09g06760	Chr9:4995997-4997377	678	225	5.62	26.02
PcGST33	pycom09g17830	Chr9:18321111-18324240	798	265	8.78	29.32
PcGST34	pycom10g14850	Chr10:18226369-18227186	660	219	5.70	25.29
PcGST35	pycom10g14860	Chr10:18230656-18231515	678	225	6.11	25.86
PcGST36	pycom10g14870	Chr10:18235274-18236268	738	245	6.42	28.46
PcGST37	pycom10g16980	Chr10:20171035-20172247	660	219	5.55	25.42
PcGST38	pycom10g19580	Chr10:22635105-22637576	1167	388	6.38	44.02
PcGST39	pycom11g26520	Chr11:29074954-29075658	483	160	5.31	18.21
PcGST40	pycom12g12570	Chr12:15027339-15028517	366	121	9.45	13.74
PcGST41	pycom13g07080	Chr13:4692196-4693096	702	233	5.88	25.66
PcGST42	pycom13g21680	Chr13:18606904-18610817	954	317	5.27	35.80
PcGST43	pycom14g19350	Chr14:20703306-20703940	429	142	7.73	16.56
PcGST44	pycom14g19760	Chr14:21009613-21012410	999	332	8.28	36.49
PcGST45	pycom15g00440	Chr15:265642-266270	366	121	7.69	13.54
PcGST46	pycom15g12040	Chr15:8191500-8192547	804	267	9.75	31.35
PcGST47	pycom15g35830	Chr15:35918255-35922360	1224	407	6.62	45.24
PcGST48	pycom15g36060	Chr15:36157385-36159383	942	313	9.06	34.82
PcGST49	pycom15g38640	Chr15:38398976-38402501	1635	544	9.30	61.98
PcGST50	pycom15g39250	Chr15:39135295-39139029	696	231	5.48	26.15
PcGST51	pycom16g07010	Chr16:4552487-4554499	624	207	5.15	22.69
PcGST52	pycom16g19480	Chr16:16579477-16580474	666	221	6.39	25.74
PcGST53	pycom16g21400	Chr16:19849382-19855412	951	316	5.62	36.15
PcGST54	pycom17g12850	Chr17:10560141-10562650	630	209	5.98	24.26
PcGST55	pycom17g26130	Chr17:24436789-24440104	798	265	8.94	29.30
PcGST56	pycom17g26990	Chr17:25250001-25251072	663	220	5.60	25.03
PcGST57	pycom17g27080	Chr17:25355973-25357297	648	215	5.50	24.54

## Data Availability

The datasets are available in the Sequence Read Archive (BioProject ID: PRJNA781276).

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
