# Peer review of "Genomic Analysis of the Glutathione S-Transferase Family in Pear (Pyrus communis) and Functional Identification of PcGST57 in Anthocyanin Accumulation"

_ijms, 2022, doi:10.3390/ijms23020746_

Round 1
Reviewer 1 Report
The manuscript contains a comprehensively study in which 57 GST genes were identified in the 'Bartlett' pear genome and divided into 10 subfamilies. There were analysed the phylogenetic evolution, syntenic relationships, the gene structure, chromosomal localization, collinearity relationship, and cis-elements of PcGST genes.
The manuscript seems interesting and appropriate as a topic for the IJMS journal.
The manuscript is well structured and clearly presented, and the results are based on adequate and suggestive tables and figures data.
I recommend a general revision by the authors for small corrections regarding some notions or terms, proper wording and a better understanding of some sentences or phrases.
Please see below just several examples/issues which must be revised.
Starting with the Abstract (see ‘Bartlett’, in Line L17), you use single quotation marks ‘…’ for the cultivar, which is correct and follows the general practice in scientific publications regarding cultivars.
But please do this in the entire paper, because the style should be used consistently throughout the manuscript. Consequently, cultivar names must be also surrounded by single quotation marks in the whole manuscript, both for Bartlett (see Lines L126, 275, 394, 402, 431, 475 for Bartlett) and other cultivars.
Before using the term 'Bartlett DH' (L101) for the first time, it would be good to explain to the reader what mean the abbreviation (DH).
Does it derive from the "double haploid (DH) plant" (Bouvier et al., 2002)?
Bouvier L. et al. Chromosome doubling of pear haploid plants and homozygosity assessment using isozyme and microsatellite markers. Euphytica 2002, 123(2), 255–262., doi:10.1023/A:1014998019674.
Check and correct punctuation, e.g.:
L 238 ‘Red Zaosu.’
L 265 ‘Zaosu,’ ‘Mansoo,’ ‘Danxiahong,’ ‘Red Clapp’s Favorite,’ and ‘Red Zaosu,’ etc.
Please revise in the manuscript some unclear statements, which require the reader to guess what the authors’ intended to mean, i.e.:
Lines L 475-477: In this study, 57 GST genes were identified in the Pyrus communis Bartlett DH genome and divided into 10 subfamilies, the gene structure, chromosomal localization, collinearity relationship, and cis-elements of PcGST genes were comprehensively analyzed.
Reviewer 2 Report
The presented article is devoted to the analysis of genes associated with anthocyanin accumulation in pear. It is a complex investigation of the glutathione S-transferase genes family. Different variants of DNA and RNA analyses were used. The authors identified 57 glutathione S-transferase genes and analyzed their nucleotide structure, localization on the chromosomes, their functions and relationships.
The article is logically structured; the conclusions are justified by detailed supplementary materials. Used methods are appropriate.
The article is written in good English.
Article can be published in present form.
Reviewer 3 Report
In this manuscript, a total of 57 GST genes were identified in the genome of the ‘Bartlett’ pear and divided into 10 subfamilies through comprehensive genomic analysis, mainly regarding the anthocyanin accumulation. The ms was well prepared and provided interesting results that could improved the breeding of pear word-wide. All our comments are in the attached file. After these changes, the ms can be accepted for publication in IJMS.

Reviewer 4 Report
Manuscript ID: ijms-1500767
After thorough review of the ms entitled “Genomic Analysis of the Glutathione S-transferase Family in Pear and Functional Identification of PcGST57 in Anthocyanin Accumulation” (Manuscript ID: ijms-1500767), this reviewer recommends it for publication after the suggested revisions.
This research investigated the mechanisms underlying the anthocyanin accumulation conferring red coloration to peels of some pear (Pyrus communis) cultivars. Since glutathione S-transferase (GST) are enzymes also regulating anthocyanin accumulation, in this work 57 PcGST genes of the pear cultivar ‘Bartlett’ were in silico identified and characterized. Further analysis by RNA sequencing (RNA-seq) showed that the amino acid glutamic acid (Glu) induced the expression of PcGST genes. Among these, PcGST57 was induced at high levels and was related to GST genes involved in anthocyanin accumulation of other plant species. Gene expression analysis showed that PcGST57 was transcribed in flesh, and other plant tissues, being related to red coloration of pear peel at different developmental stages. Overexpression of PcGST57 increased anthocyanin accumulation in pear peel, while silencing by virus-induced gene silencing (VIGS) inhibited the expression of PcGST57 and decreased the anthocyanin content. Authors provided convincing evidences that PcGST57 can be a candidate gene involved in the accumulation of anthocyanins in the red pear.
Despite their findings, Authors have to exactly indicate the novelty of this study compared to other works on plant GST. Therefore, this reviewer suggests revisions of the text, also by improving English language and by removing inaccuracies/repetitions.
Further suggestions are listed below:
TITLE
L3: insert “(Pyrus communis)” after “Pear” in “Genomic Analysis of the Glutathione S-transferase Family in Pear and Functional Identification of PcGST57 in Anthocyanin Accumulation”
- INTRO
L47: change “converted” to “produced starting”
L48: complete the biosynthetic pathway, insert “C4H, 4CL” after “PAL”
L49: complete the biosynthetic pathway, insert “F3’H” after “F3H”
L57: remove “reported to be”
L56: insert “(Eustoma grandiflorum)” after “petals”
L64: change “Glutathione-S transferase” to “Glutathione S-transferases”, change “is” to “are”
L67: remove comma in “biotic,”
L83: consider changing “systemic” to “systematic”
- RESULTS
L116: see L105 and 419, use MEGA X instead of MEGS 6.0
L125: change “were” to “was”
L 114: in Figure 1, remove hyphen in “GST-M” used on phylogenetic tree
L153-160 and L161-168: consider inverting these two paragraph each other, so describing before Figure 4A and then Figure 4B
L174-176 and L477: use italics for “cis”
L188: use “glutamate” or “glutamic acid”, not both of them
L190-191: here, reference(s) needed
L196: clarify “partial” for regulatory genes
L200: consider changing “Figure 7” to “Figure 6”
L203: remove repetition of “fold change”, then insert “at 72 h”, in Figure 6 the expression scale ranges from -3 to +3, how can the fold change of 18 be shown? Please fix
L237: insert “Figure 8” after ‘Red Clapp’s Favorite’, move full stop to the end (‘Red Clapp’s Favorite.’)
L243: the first time, specify the acronym DAF as “day after full blossom” (see L396-7)
L245: move Figure 8B after “petal”
L264: remove repetition of “different pear cultivars”
L268-269: move this sentence to heading in L264, after “Figure 8. The transcript patterns of PcGST57.”
L269: fix typing error in “PcTublin”
L281: remove “OE-EV”, this is not in Figure 9
L281, 283, 285, 291, 294: check and fix the incorrect references to the letters of Figure 9
L287-288: the first time, specify the acronyms TRV and VIGS
L292: insert “Figure 9F” after “peels”
L302: change to “PcTubulin was used as the internal control for gene expression analysis”
- DISCUSSION
L306: change “in” to “of”
L313: add “species” after “plants”
L320-321: change “event” to “events”
L381: please check and fix, change “Figure 6” to “Figure 5”
- M&M
L454: insert “phenol” before “determination”
L462: after conversion, change “rpm” to “RCF or g”
- CONCLUSIONS
L481: remove “could”, and change to “improved and decreased”
L482: consider changing “was” to “is”
